# Dispensing of Non-Prescribed Antibiotics from Community Pharmacies of Pakistan: A Cross-Sectional Survey of Pharmacy Staff’s Opinion

**DOI:** 10.3390/antibiotics10050482

**Published:** 2021-04-22

**Authors:** Muhammad Majid Aziz, Fatima Haider, Muhammad Fawad Rasool, Furqan Khurshid Hashmi, Sadia Bahsir, Pengchao Li, Mingyue Zhao, Thamir M. Alshammary, Yu Fang

**Affiliations:** 1Department of Pharmacy Administration and Clinical Pharmacy, School of Pharmacy, Xi’an Jiaotong University, Xi’an 710061, China; pharmajid82@yahoo.com (M.M.A.); lipengchao1996@stu.xjtu.edu.cn (P.L.); mingyue0204@xjtu.edu.cn (M.Z.); 2Center for Drug Safety and Policy Research, Xi’an Jiaotong University, Xi’an 710061, China; 3Global Health Institute, Xi’an Jiaotong University, Xian, Xi’an 710049, China; 4Shaanxi Center for Health Reform and Development Research, Xi’an 710061, China; 5Ghazi Medical College, Jampur Rd, Dera Ghazi Khan, Punjab 32200, Pakistan; dr-f.haider14512@yahoo.com; 6Departmentof Pharmacy Practice, Faculty of Pharmacy, Bahauddin Zakariya University Multan, Punjab 60800, Pakistan; fawadrasool@bzu.edu.pk; 7University College of Pharmacy, University of the Punjab, Lahore 54590, Pakistan; furqan.pharmacy@pu.edu.pk; 8Department of Pathology, Quaid e Azam Medical College, Punjab 63100, Pakistan; dr-sadiabashir84@yahoo.com; 9College of Pharmacy, Riyadh Elm University, Riyadh 11564, Saudi Arabia; Thamer.alshammary@gmail.com

**Keywords:** community pharmacies, dispensing, antibiotics, non-prescribed, knowledge, attitude, practices

## Abstract

Community pharmacies are the main channel of antibiotics distribution. We aimed to analyze the dispensing of non-prescribed antibiotics and knowledge of pharmacy staff. We conducted a cross-sectional study in Punjab, Pakistan between December 2017 and March 2018. A self-administered, structured, pretested, and validated bilingual questionnaire was used, and we used chi-square tests in the statistical analysis. A total of 573 (91.7%) pharmacy retailers responded to the survey; 44.0% were aged 31–40 years and all were men. Approximately 81.5% of participants declared that dispensing non-prescribed antibiotics is a common practice in community pharmacies, and 51.1% considered themselves to be authorized to dispense these drugs; 69.3% believed this a contributing factor to antimicrobial resistance. Most (79.1%) respondents believed that this practice promotes irrational antibiotics use, and half (52.2%) considered antimicrobial resistance to be a public health issue. Only 34.5% of respondents reported recommending that patients consult with a doctor prior to using antibiotics, and 61.8% perceived that their dispensing practices reduce patients’ economic burden. Approximately 44.9% of pharmacy retailers stated that they have proper knowledge about antibiotics use. Nitroimidazole was the main class of antibiotic dispensed without a prescription. Dispensing of injectable and broad-spectrum antibiotics can be potential threat for infection cure. Poor knowledge of staff is associated with dispensing of non-prescribed antibiotics. This inappropriate practice must be addressed immediately.

## 1. Introduction

Community pharmacies are consideredas a main source of antibiotics’ distribution, worldwide [1]. Dispensing of non-prescribed antibiotics (DoNA) through these channels is a common phenomenon [2,3,4,5,6,7,8]. DoNA promotes irrational medication in the community [3,4]. The World Health Organization reports that unnecessary and inappropriate medication with antibiotics is amplifying antimicrobial resistance (AMR) [5]. Pakistan has a high AMR owing to extensive misuse of antibiotics [6]. Currently, extensively drug-resistant typhoid fever has created an epidemic situation in Pakistan [6,7]. Multidrug resistance is also high in the country (>25–100%), and AMR is increasing owing to increased sales of antibiotics [8].

Pakistan ranks third for antibiotics consumption among low- and middle-income countries (LMICs). Antibiotics consumption expressed in defined daily doses (DDD) increased 65% during 2000–2015 [9]. It is estimated that 35,000 patients per day use antibiotics in Pakistan [10]. More than half of rural residents use antibiotics without a prescription [11,12], and more than 35% of antibiotics are sold by urban pharmacies without a valid prescription [13]. Every Pakistani has easy access to antibiotics that are sold as over-the-counter (OTC) medications [5,14]. A survey of pharmacies found that antibiotics without prescription are the most frequently sold among all prescription-only-medicines (PoM) [15]. However, the sale of non-prescribed antibiotics is completely prohibited by the national drug policy (NDP) of Pakistan. The NDPemphasizes that all types of antibiotics should sold by pharmacies only with a valid prescription from a registered doctor [16].

Approximately 80,000 community pharmacies are integrated within the health care system of Pakistan [17], and it is estimated that 80% of medicines are distributed through community pharmacies [18]. These pharmacies have the potential to improve the rational use of antibiotics in the community [5,14,18]. Community pharmacists are ideally positioned as antibiotic stewards whose dispensing practices can help to reduce AMR [19,20,21]. Still, no studies have been conducted to evaluate the knowledge, attitudes, and practices regarding DoNA among community pharmacists in Pakistan. Thus, we performed the present study to fill this research gap.

## 2. Results

A total of 573 (91.7%) staff members of pharmacies participated in this study. Among them, 5.9% completed the survey in face-to-face interviews. Approximately 4.4% of staff refused to participate and 3.8% returned partially completed questionnaires; these questionnaires were excluded from the final results.

### 2.1. Participant Demographics 

The mean age of respondents was 34.9 ± 2.6 years; among them, 44.0% were aged 31–40 years and all were men. Most pharmacy retailers (81.5%) had a non-professional education; approximately half (51.3%) had working experience of more than 10 years, and half (51.3%) declared their status was an employee of the pharmacy. More than half of the participants filled up to 50 prescriptions per day (54.8%) anddispensed up to 20 antibiotics per day (50.1%); less than half dispensed fewer than 10 non-prescribed antibiotics per day (47.7%) (Table 1).

### 2.2. Knowledge and Attitudes about DoNA

Most survey respondents (81.5%) stated that DoNA is a common practice in community pharmacies. Only 212 (37.0%) believed that dispensing PoM without a prescription causes health issues (Table 2).

### 2.3. Practices of DoNA

Among respondents, 366 (63.9%) reported that a main reason for DoNA was patients’ inability to afford a doctor’ s consultation fee. Most (80.6%) pharmacy retailers dispensed non-prescribed antibiotics in oral dosage form (Table 3).

Most (72%) pharmacy staff dispensed non-prescribed antibiotics for the treatment of gastrointestinal tract infections; 69.1% dispensed these drugs for respiratory tract infections and 43.9% for urinary tract infections (Figure 1).

At the time of dispensing non-prescribed antibiotics, 87.1% counsel the patients for medication adherence and compliance. Only 46.2% of study participants don’t dispense non-prescribed antibiotics for children as given in Table 4.

### 2.4. Suggestions to Stop DoNA

Among those surveyed, 397 (69.2%) pharmacy retailers suggested that continuous training about AMR can help to stop DoNA. Most respondents (79.9%) suggested a role for pharmacies in patient education on the outcomes of DoNA and AMR (Table 5).

## 3. Discussion

The present study was the first to evaluate knowledge and attitudes among pharmacy retailers and practices of DoNA in community pharmacies of Pakistan. Similar to other countries with developing health care systems [1,2,4,22,23,24,25], the current study revealed the high incidence of DoNA in community pharmacies. All study participants divulged that DoNA was their common practice. This study revealed poor knowledge among drug sellers. Only half of respondents believed that dispensing of PoM without a prescription causes health issues. A main reason behind the poor knowledge of staff is the inadequate supervision of pharmacies in Pakistan [14]. Similarly, nearly one-quarter of pharmacy retailers believed that DoNA is not a factor contributing to AMR. This was uncovered in a previous study reporting that pharmacists in Punjab have similar misconceptions. Some community pharmacists in Punjab perceive that irrational antibiotics use does not lead to AMR [26]. This pattern is observed in other LMICs as well as developed countries [27,28]. Only 55.5% of participants in this study were in favor of stopping DoNA in their pharmacies and only one-third reported that they recommend that patients consult with a doctor before using antibiotics. Collaboration between community pharmacies and general practitioners is an excellent model to help prevent AMR [29]. 

It is well known that inappropriate antibiotics use increases therapeutic costs [30]. However, the participants in this study had varying viewpoints in this regard. Similar to pharmacists in Egypt, pharmacy staff in Pakistan considered that DoNAprovides economic benefit to patients [31]. This misconception among respondents was coupled with unawareness of applicable legislation, which is an important aspect that enhances compliance with regulations. In this study, we showed that pharmacy retailers had insufficient knowledge about regulations regarding the sale of antibiotics without a prescription. Similar to a study from Saudi Arabia [4], most participants believed that they were authorized to engage in DoNA, which revealed their lack of awareness of the NDP [16].

A main reason for selling antibiotics without a prescription was associated with affordability for patients. Similar findings have been reported in Ethiopia, where non-prescribed antibiotics are frequently used by people with few economic resources [32]. Health-seeking behavior is also an important reason for DoNA in community pharmacies. Pharmacy staff in our study revealed that patients visit a physician only for serious infections, which resembled findings of a previous study [4] regarding health-seeking behaviors of the population.

Similar to pharmacists in Thailand, pharmacy staff in Pakistan considered themselves sufficiently competent to dispense antibiotics without a physician’s prescription [33]. However, previous reports from community pharmacies in Pakistan have indicated improper dispensing of antibiotics, with sub-therapeutic doses [14,34].

Nitroimidazole was the class of antibiotics most frequently sold without a prescription. OTC sales of this antibiotics class have been reported in several previous studies from Pakistan and Zambia [15,24]. As in Egyptian pharmacies, we also identified OTC sales of penicillins, quinolones, and macrolides [30]. Similar to previous findings from Cameroon, our study also revealed the use of injectable antibiotics without a prescription [35]. As in studies from China and Mongolia, we revealed the use of non-prescribed antibiotics for children [36,37]. Moreover, use of non-prescribed antibiotics is also common among pregnant women in Pakistan [15]. The high DoNA reveals the poor implementation of NDP.

However, as recommended in previous research [25,28,32,38], participants in this study suggested adequate training of staff in community pharmacies and structured public awareness campaigns about AMR. Holding awareness seminars at chemist and druggist association meetings was suggested, to improve rational antibiotics use. Additionally, mechanisms of regulation and policy interventions should be further strengthened [39]. According to our study results, staff training, strong vigilance with respect to policy implementation, and public health awareness are recommended, to help in stemming DoNA.

## 4. Methods

### 4.1. Study Setting

Study sites were community pharmacies located in three divisions (Bahawalpur, Dera Ghazi Khan, and Multan, known as “south Punjab”) of Punjab, Pakistan. The total population in this area is 34,747,064 [40]. Whereas total 22,319 community pharmacies are located in Punjab province [41].

### 4.2. Sampling and Pharmacy Selection

A sample size of 625 community pharmacies was calculated using Raosoft online software to obtain a total of 10,731 pharmacies with a valid license for drug sales [42]. For the total number of pharmacies, a list of registered pharmacies was taken from the department of health in each district and summarized [43]. For sample size calculation, we considered the response distribution (50%), confidence interval (99%), margin of error (5%), and response rate in a pilot study. Pharmacies were selected using a random sampling method, and randomization was performed with systematic selection. 

### 4.3. Participant Inclusion Criteria

A staff member from each selected pharmacy who consented to participate was included in this study. All respondents participated on a voluntary basis. Only staff members aged ≥18 years and with at least 5 years’ experience were included. Prior to completing the survey, participants were briefed about the purpose of the study and were requested to answer based on their own perceptions and to refrain from discussing the survey with other colleagues. 

### 4.4. Study Instrument

A questionnaire was developed on the basis of previous studies [2,4,14,15,22,23,24,25,32,33,44,45], after a very extensive literature review (February to November 2017). Based on the literature, we extracted a pool of 38 questions. The face validity, relevance, and clarity of the content were assessed by experts in community pharmacy. The instrument was modified per the advice of experts, and unnecessary items were eliminated. The content of 35 items was further validated by three community pharmacists; the rating method was applied for validation. Ratings for each item were obtained using an item-objective congruence (IOC) score method, with 1 = clearly not measured, 0 = unclear content, and 1 = clearly measured. An average score of IOC > 0.5 was considered to indicate good content validity [46]. Urdu (the national language) translation of the questionnaire was also completed by three language experts, with forward and back translation. To ensure the accuracy of translation, we used a inter-rater reliability method. Agreement between two or more raters was considered good agreement.

To further ensure the clarity of the translation and modifications to the survey, face and content validity was evaluated using a cognitive interview method among 10 randomly selected participants from the included pharmacies [47,48,49,50]. The 31-item questionnaire was finalized. In the cognitive interviews, each item was rated as relevant, understood, appropriate, not difficult, and able to be correctly interpreted by >80% of respondents [51]. The reliability and internal consistency of the study instrument was then assessed in a pilot study. Reliability was also assessed using the Cronbach’s alpha value. An acceptable value of 0.7 was used as our benchmark. The Cronbach’s alpha of the knowledge, attitudes, practices, and suggestion portions was 0.81, 0.79, 0.75, and 0.83, respectively, thereby ensuring a reasonable level of reliability and internal consistency [3]. The Cronbach’s alpha value for the entire questionnaire was 0.752. Thus, the final questionnaire (Appendix A) comprised 31 questions and 4 parts: the first part included 8 open-ended questions on respondents’ demographic characteristics. Open-ended questions were preferred because of probable diversity in the sample. The second part consisted of 10 questions addressing knowledge and attitudes, with response options of “yes”, “no”, and “don’t know”. The third portion addressing practices consisted of 11 questions and the final part had 2 questions regarding suggestions for halting DoNA.

### 4.5. Pilot Study

We conducted a pilot study among staff members of 30 conveniently selected community pharmacy to evaluate the length of the survey and clarity of the questions. The results of the pilot study were not included in the final results.

### 4.6. Data Collection

For this cross-sectional study, the data were collected between December 2017 and March 2018. The self-administered, bilingual (Urdu and English) questionnaires were distributed in the selected pharmacies. Data collection was completed over three visits. During the first visit, data collectors distributed the questionnaires to pharmacy staff. After 5 days, the data collectors visited the pharmacies to collect completed questionnaires. To ensure the maximum participation of staff, the pharmacies were visited again 3 days after the second visit. During the third visit, some pharmacy staff members were interested in participating in the study but did not have time to complete the questionnaire owing to their busy schedules. Therefore, the data collectors conducted face-to-face interviews among participants who agreed to be interviewed. Data entry was double checked by two investigators. 

### 4.7. Data Analysis

The results were analyzed by calculating the frequency, percentage, and standard deviation. A 6-point Likert scale (always, very often, sometimes, rarely, very rarely and never, corresponding to points from 1 to 6) was used to assess the cumulative values for DoNA practices. To understand the relationship between participants’ characteristics and DoNA practices, we used the chi-square test. SPSS version 18.0 was used for the statistical analyses (SPSS Inc., Chicago, IL, USA).

### 4.8. Ethics Approval

The study design and protocols were approved by the Centre for Drug Safety and Policy Research at the School of Pharmacy after formal approval by the ethical review committee of Xi’an Jiaotong University (Ref #MR102-15/Phar) and Pharmacy Research Ethics Committee at The Islamia University Bahawalpur, Pakistan (Ref #67-2015/PREC). The reference number (CDSP-16-PHD1-P4) was obtained before conducting the study. All participants were well informed of the study purpose and written and verbal consent was obtained from each participant. Participants’ identity was anonymized, and identification numbers were used in data collection, monitoring, and entry. 

## 5. Conclusions

Our survey showed the dispensing of injectable and broad-spectrumantibiotics without prescription. That can be prospective potential threat for infection cure. The knowledge about AMR was inadequate among community pharmacy staff. Moreover, awareness of the regulations and policies regarding OTC sales of antibiotics were poor. This lack of awareness potentiates the high rates of DoNA. To combat rising AMR, Pakistan must develop multifaceted and comprehensive programs to enhance knowledge among drug sellers, and proper execution of the NDP is needed.

## 6. Limitations

First: this study was based on a self-administered questionnaire; the results obtained are purely on the basis of feedback from pharmacy staff. Therefore, we used a six-point Likert scale to assess practices and a three-point scale for attitudes, which can help to reduce inherent biases. Moreover, we collected data in three steps, and participation was purely voluntarily. Second, only one staff member from each pharmacy was included. Including multiple participants from a single pharmacy may have affected the final results. Third, the current study was conducted in the southern region of Punjab, Pakistan; the results may vary from region to region. However, the participant diversity with respect to age, educational level, and practices justifies the study results as baseline and representative data. Fourth, all study participants were men, owing to specific sociocultural traits of the study setting. Fifth, this study represents the views of drug sellers with more than fiveyears’ work experience. All study participants were men; the proportion of female staff is extremely small in community pharmacies in male-dominated societies. Finally, to determine actual practices, the educational background of staff was not used as an inclusion criterion because most community pharmacies in the country employ staff without a professional education.

## Figures and Tables

**Figure 1 antibiotics-10-00482-f001:**
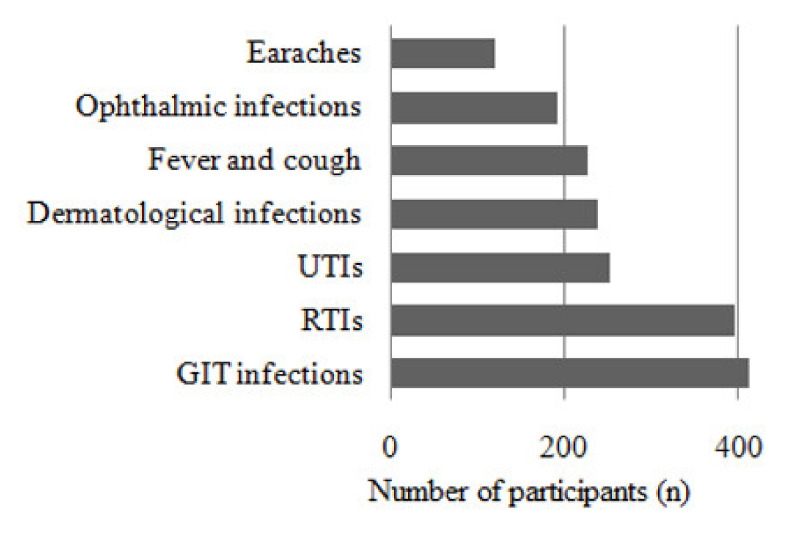
Diseases mentioned by participants for which non prescribed antibiotic are dispensed.

**Table 1 antibiotics-10-00482-t001:** Characteristics of study participants.

Characteristics	Range/Groups	Frequency (%)
Age (Years)	30 or below	159 (27.7)
	31–40	252 (44.0)
	41–50	137 (23.9)
	51 or above	25 (4.4)
Mean age ± SD = 34.9 ± 2.6		
Gender	Male	573 (100.0)
	Female	0 (0.0)
Education	Formal education	467 (81.5)
	Professional education	106 (18.5)
Experience (Years)	5–10	279 (48.7)
	11–15	222 (38.7)
	More than 15	72 (12.6)
Status in pharmacy	Employee	294 (51.3)
	Manager	41 (7.2)
	Proprietor	238 (41.5)
Prescriptions (all types of medicines) per day	Less or 50	314 (54.8)
	51–100	176 (30.7)
	More than 100	83 (14.5)
Total antibiotics dispensed per day(Prescribed + Non prescribed)	Less or 20	287 (50.1)
	21–40	197 (34.4)
	More than 40	89 (15.5)
DoNA per day	Less or 10	273 (47.7)
	11–20	199 (34.7)
	More than 20	101 (17.6)

**Table 2 antibiotics-10-00482-t002:** Knowledge and attitude of participants about DoNA.

Measures	Yes n (%)	No n (%)	Don’t Know n (%)
Community pharmacies in Pakistan are authorized for DoNA	293 (51.1)	109 (19.1)	171(29.8)
DoNA from community pharmacies of Pakistan is common practice	467 (81.5)	22(3.8)	84(14.7)
Dispensing of PoM without prescription causes health issues	212 (37.0)	289 (50.4)	72(12.6)
DoNA contributes to the development of AMR	397 (69.3)	128 (22.3)	48(8.4)
AMR has become a challenging issue of public health	299 (52.2)	115 (20.1)	159(27.7)
DoNA is a cause of irrational use of antibiotics by patients	453 (79.1)	104 (18.1)	16(2.8)
Pharmacy retailers should stop DoNA	318 (55.5)	222 (38.7)	33(5.8)
I recommend doctor’s consultation to the patients before use of any antibiotic	198 (34.5)	375 (65.4)	^__________ *^
On my refusal to DoNA, patient will try to get antibiotics from another pharmacy	503 (87.8)	62(10.8)	8(1.4)
DoNA reduces the economic burden of patients	354 (61.8)	88 (15.4)	131(22.8)

* Question according to its nature includes only two answering options: Yes or No.

**Table 3 antibiotics-10-00482-t003:** Dispensing of non-prescribed antibiotics.

Measures	Items	Frequency (%)
Reasons for DoNA	Pharmacy retailers have proper knowledge about use of antibiotics	257 (44.9)
	Patients visit physicians only for serious infections	317 (55.3)
	Increases the profit and sales	97 (16.9)
	Lack of patients’ affordability for consultation fee of doctors	366 (63.9)
	Fear of customer loss	93 (16.2)
	Unawareness of pharmacy staff toward policy and regulations	107 (18.7)
Classes of commonly dispensed non prescribed antibiotic	Nitroimidazole	411 (71.7)
	Penicillins	382 (66.7)
	Quinolones	337 (58.8)
	Cephalosporins	207 (36.1)
	Tetracycline	116 (20.2)
	Macrolides	91 (15.9)
Administration form of commonly dispensed non prescribed antibiotic	Oral	462 (80.6)
	Topical cream/ointment	294 (51.3)
	Ophthalmic ointment/drops	271 (47.3)
	Ear drops	211 (36.8)
	Parenteral/Injectables	17 (2.9)

**Table 4 antibiotics-10-00482-t004:** Association between DoNA practices and characteristics of participants.

Measures	Score	*p*-Values	
	Mean ± SD	Age	Education
Before DoNA, I inquire thedrug allergies to patients	3.7 ± 0.10	0.050	0.572
Before DoNA, I inquire about the renal condition of patients	2.5 ± 0.06	0.048	0.162
Before DoNA, I ask the patient about other disease/s or therapies	4.9 ± 0.13	0.573	0.201
At the time of DoNA, I inform the patients about possible side effects	3.0 ± 0.08	0.047	0.411
At the time of DoNA, I counsel the patients for medication adherence and compliance	5.2 ± 0.14	0.172	0.070
I don’t dispense non-prescribed antibiotics for children	2.7 ± 0.07	0.701	0.090
I don’t dispense non-prescribed antibiotics for pregnant women	3.0 ± 0.08	0.849	0.385

*p*-values < 0.05 = statistically significant; Mean value and score weighting from 6-point Likert scale (1 = never, 2 = very rarely, 3 = rarely, 4 = some time, 5 = very often, 6 = always).

**Table 5 antibiotics-10-00482-t005:** Suggestions from participants to halt DoNA.

Measures	Items	Frequency (%)
To control DoNA from community pharmacies	Pharmacy retailers should penalized	92 (16.1)
	Vigilance of pharmacies by drug inspectors should be more strong	78 (13.6)
	Pharmacy retailers need continuous trainings	397 (69.2)
	Other e.g., reduction ofdoctors’ consultation fee, increase in free health facilities	17 (2.9)
Role of pharmacy retailers to halt DoNA	Educate patients about the outcome and AMR due DoNA	458 (79.9)
	Educate and train the staff of pharmacy about the outcome and AMR due DoNA	347 (60.6)
	Arranging awareness seminars in Chemist and Druggist Associations’ meetings	272 (47.5)
	Arrange public awareness campaigns	167 (29.1)

## Data Availability

The data presented in this study are available on request from the corresponding author.

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
