# Peer review of "Dispensing of Non-Prescribed Antibiotics from Community Pharmacies of Pakistan: A Cross-Sectional Survey of Pharmacy Staff’s Opinion"

_antibiotics, 2021, doi:10.3390/antibiotics10050482_

Round 1

Reviewer 1 Report

This paper describes a very critical issue in antibiotic stewardship and I agree that an intervention needs to be done as soon as possible. It is devastating to know that pharmacy employees without educational background are allowed to dispense potential harmful medications. 

The order of the manuscript is confusing. Method should be after introduction followed by results then discussion.

Specific comments per line:

Line 46: the range of MDR seeps to be huge. Is is only based on 1 reference?

Line 48: Antibiotic use increased to 65% from what?

Lines 52-53 is confusing as the statement is contradictory: OTC antibiotics but then prescription-only medicines

Lines 54-57: Should include in the discussion why pharmacies are still dispensing antibiotics without prescription despite the national drug policy.

Line 67: 573 pharmacy or pharmacy staff?

Table 4 is confusing. Consider revising or reformatting

Line 214: Re the 30 participants for the pilot study- were they pharmacy staff?

Author Response

Comment

This paper describes a very critical issue in antibiotic stewardship and I agree that an intervention needs to be done as soon as possible. It is devastating to know that pharmacy employees without educational background are allowed to dispense potential harmful medications. 

Response

We are very grateful to you for providing the professional views and constructive comments

Comment

The order of the manuscript is confusing. Method should be after introduction followed by results then discussion.

Response

In author guidelines (https://www.mdpi.com/journal/antibiotics/instructions). The order is Introduction, Results, Discussion, Materials and Methods, Conclusions (optional).

So this is journal requirement

Specific comments per line:

Comment

Line 46: the range of MDR seeps to be huge. Is is only based on 1 reference?

Response

This is "Situation analysis report on antimicrobial resistance in Pakistan". A comprehensive and first report on AMR

Comment

Line 48: Antibiotic use increased to 65% from what?

Response

Sorry for poor English of this sentence. Its due to English expression, so it was interpreted different. Now it's  corrected and explained  as

"Antibiotics consumption expressed in defined daily doses (DDD) increased 65% during 2000–2015. (21.1–34.8 billion DDDs), and the antibiotic consumption rate increased 39% (11.3–15.7 DDDs per 1,000 inhabitants per day)"

We hope now it's understandable now.

Comment

Lines 52-53 is confusing as the statement is contradictory: OTC antibiotics but then prescription-only medicines

Response

Once again thanks for highlighting this mistake. Its corrected as

"A survey of pharmacies found that antibiotics without prescription are the most frequently sold among all prescription-only-medicines (PoM)"

Comment

Lines 54-57: Should include in the discussion why pharmacies are still dispensing antibiotics without prescription despite the national drug policy.

Response

Included in the discussion "The high rate of DoNA reveals the poor implementation of NDP".

Comment

Line 67: 573 pharmacy or pharmacy staff?

Response

Its corrected and written as "A total of 573 (91.7%) staff members of pharmacies participated"

Comment

Table 4 is confusing. Consider revising or reformatting

Response

Corrected as advice and caption is added that will make the table self explanatory. 

Comment

Line 214: Re the 30 participants for the pilot study- were they pharmacy staff?

Response

Its corrected and written as

We conducted a pilot study among staff members of 30 conveniently selected community pharmacy to evaluate the length of the survey and clarity of the questions.

Thanks and Best Regards

Reviewer 2 Report

Interesting paper, accessible, with elevated pertinence in regard with antibiotic resistance. Pleasant to read.

Remark 1: geographical location is well described, but you don't give an idea of the population of Punjab and of the sample cohort covered by your investigated pharmacies. These two numbers would add value to your manuscript.

Remark 2: Table 4. Association between DoNA practices and characteristics of participants.

Mean score (Likert scale): OK, but it should be easier for the reader if you add the significance of the 6-levels Likert scale as a note under the Table 4 (never,..., always) in place of the section Methods (I had to seek fastidiously this information).

Percentages: is it well the percentage of Yes answer? Specify it in the table or in the text - results section.

Probabilities: is it well Pearson correlation coefficient between Liket scores and other variables? As Liket scores are not gaussian, Pearson correlation is not adequate. I do'nt understand these probabilities: has also to be clarified in a note under the table.

In conclusion, this table 4 and the section Results in relationship with this table should be improved. Look also if it would not be better to mention in this table only Liket scores and percentages Yes/No: many times, a short table attracts more attention than a table with too many data. It seems that this table is at the central to your observations.

Author Response

Interesting paper, accessible, with elevated pertinence in regard with antibiotic resistance. Pleasant to read.

Response

We are very grateful to you for providing the professional views and constructive comments

Remark 1: geographical location is well described, but you don't give an idea of the population of Punjab and of the sample cohort covered by your investigated pharmacies. These two numbers would add value to your manuscript.

Response

Study sites were community pharmacies located in three divisions(Bahawalpur, Dera Ghazi Khan, and Multan, known as "south Punjab") of Punjab, Pakistan. The total population in this area is 34,747,064 . Whereas total 22,319 community pharmacies are located in Punjab.

Remark 2: Table 4. Association between DoNA practices and characteristics of participants.

Mean score (Likert scale): OK, but it should be easier for the reader if you add the significance of the 6-levels Likert scale as a note under the Table 4 (never,..., always) in place of the section Methods (I had to seek fastidiously this information).

Percentages: is it well the percentage of Yes answer? Specify it in the table or in the text - results section.

Probabilities: is it well Pearson correlation coefficient between Liket scores and other variables? As Liket scores are not gaussian, Pearson correlation is not adequate. I do'nt understand these probabilities: has also to be clarified in a note under the table.

In conclusion, this table 4 and the section Results in relationship with this table should be improved. Look also if it would not be better to mention in this table only Liket scores and percentages Yes/No: many times, a short table attracts more attention than a table with too many data. It seems that this table is at the central to your observations.

Response

Captions related to likert scale is added

The table is shorten as advise.

Hope this new format will be easily under stable to readers

Thanks and Best Regards

Reviewer 3 Report

This manuscript is, " Dispensing of non-prescribed antibiotics from community 2 pharmacies of Pakistan: a cross-sectional survey of pharmacy 3 staff's opinion  " by  Muhammad Majid Aziz. This study is a cross-section study majorly investigating the concepts and knowledge about the dispensing of non-prescribed antibiotics offered by the pharmacy retailers in Punjab, Pakistan. These author indicated that nitroimidazole was the leading class of antibiotic dispensed without a prescription and poor knowledge among pharmacy staff is associated with dispensing of non-prescribed antibiotics. However, two leading concerns were discovered as the following:

First, my leading concern was the novelty, because numerous reports in varied areas and countries have discussed with crucial association of poor pharmacy knowledge and the dispensing of non-prescribed antibiotics.

Second, this relationship was evidenced in your study area. This finding is useful for establishing the policy of public health in your community, but it is inappropriate for publications in this international journal.

Author Response

This manuscript is, " Dispensing of non-prescribed antibiotics from community 2 pharmacies of Pakistan: a cross-sectional survey of pharmacy 3 staff's opinion  " by  Muhammad Majid Aziz. This study is a cross-section study majorly investigating the concepts and knowledge about the dispensing of non-prescribed antibiotics offered by the pharmacy retailers in Punjab, Pakistan. These author indicated that nitroimidazole was the leading class of antibiotic dispensed without a prescription and poor knowledge among pharmacy staff is associated with dispensing of non-prescribed antibiotics. However, two leading concerns were discovered as the following:

Comments

First, my leading concern was the novelty, because numerous reports in varied areas and countries have discussed with crucial association of poor pharmacy knowledge and the dispensing of non-prescribed antibiotics.

Response

We are very grateful to you for providing the professional views and constructive comments.

As there is the concern of novelty, this manuscript adds few points in the scientific literature rather than knowledge and attitude analysis of community pharmacy's staff

  1. In this study we found that how may antibiotics are dispensed per day (Total antibiotics dispensed per day) as given in table1.
  2. This study also reveals the number on antibiotics that are dispensed without prescription as given in table1.
  3. Most sold antibiotics without prescription are used in oral dosage form. moreover use of Topical cream/ ointment and injetables is also found in this study (as given in table 2). So dosage form of antibiotics (without prescription) is also found.
  4. Most commonly used antibiotic without prescription is Nitroimidazole but dispensing of some broad sprectrum antiobiotics like Quinolones is also revealed by this study (as given in table2). In this way this study can be helpful to forecast the development of AMR on the basis of their irrational use.  
  5. This study also found the use of antibiotic according to body system like in GIT infections most of non prescribed antibiotics are used (as given in fig).
  6. Reasons behind the malpractice are explored first time, as least reported in literature (as given in table 2).
  7. This study also suggest the possibilities of intervention and utilization of community pharmacies' potential to halt AMR ( Table 5). These suggestions can be helpful for other developing countries and can contribute in making a global intervention strategy.

So a comprehensive tool (that explores many areas was developed) is also a scientific novelty.

Comments

Second, this relationship was evidenced in your study area. This finding is useful for establishing the policy of public health in your community, but it is inappropriate for publications in this international journal

Response

Once again , We are very grateful to you for providing the professional views

According to WHO’s strategy on research for health, approved by the 63rd World

Health Assembly in May 2010,

" policies and practices in support of health worldwide should be grounded in the best scientific knowledge"

So this paper can helpful for evidence based practices and mostly of international and prestigious  journal are  providing scientific background  to evidence base practices. I hope this manuscripts with WHO frame of strategy on research for health also suits to this international Journal. 

Thanks and Best Regards

Reviewer 4 Report

Dear authors

The manuscript is well structured, needs adjustment.

I believe that editing and rewriting  same part of the text is necessary to make the manuscript easily readable, but this effort would be well worth it as the authors have included a great deal of information.

However, I have following comments that should be addressed:

In the introduction part first row need to be rewrite.

The article need to be double check by an native English speaker.

The conclusion need to focus on clinical improvement of patients treatment, and to introduce in clinical use.  The conclusion part need more explanation and clear points.

  • Please recheck the References order

Author Response

Comments

The manuscript is well structured, needs adjustment.

Response

We are very grateful to you for providing the professional views and constructive comments

Comments

I believe that editing and rewriting  same part of the text is necessary to make the manuscript easily readable, but this effort would be well worth it as the authors have included a great deal of information.

Response

Modified as advised. The changes are highlighted in tracked change copy

However, I have following comments that should be addressed:

Comments

In the introduction part first row need to be rewrite.

Response

Modified as advised. The changes are highlighted in tracked change copy

Comments

The article need to be double check by an native English speaker.

Response

Undergone the English editing and tried to improve the language.

Comments

The conclusion need to focus on clinical improvement of patients treatment, and to introduce in clinical use.  The conclusion part need more explanation and clear points.

Response

Modified as advised.

Comments

Please recheck the References order

Response

All the references are rearranged.

Thanks and Best Regards

Round 2

Reviewer 3 Report

In the revised manuscript, authors have discovered that poor knowledge among pharmacy staff is associated with dispensing of non-prescribed antibiotics. However, my leading two concerns dose not improved substantially 

First, my leading concern was the novelty, because numerous reports in varied areas and countries have discussed with crucial association of poor pharmacy knowledge and the dispensing of non-prescribed antibiotics.

Second, this relationship was evidenced in your study area. This finding is useful for establishing the policy of public health in your community, but it is inappropriate for publications in this international journal.